# Macrosymbionts of starfish *Echinaster luzonicus* (Gray, 1840) in the waters of a volcanic western Pacific island

Li-Chun Tseng[1]ᴼ, Parinya Limviriyakul[2]ᴼ, Jiang-Shiou Hwang[1,3,4]*

1 Institute of Marine Biology, National Taiwan Ocean University, Keelung, Taiwan, 2 Faculty of Fisheries, Department of Marine Science, Kasetsart University, Bangkok, Thailand, 3 Center of Excellence for Ocean Engineering, National Taiwan Ocean University, Keelung, Taiwan, 4 Center of Excellence for the Oceans, National Taiwan Ocean University, Keelung, Taiwan

ᴼ These authors contributed equally to this work.
* jshwang@mail.ntou.edu.tw

**Data Availability Statement:** All relevant data pertaining to this study are included in this published article and its supplementary information files.

## Abstract

During an investigation program of faunal diversity in the shallow reef zone of the active volcanic island off northeastern Taiwan in July and September 2020, numerous individuals of the starfish *Echinaster luzonicus* (Gray, 1840) were found, and some individuals were found with associated symbionts. Starfish sampling in the 150-m coral reef zone was undertaken at a depth of 8 m through scuba diving. For each type of potential macrosymbiont, both the dorsal and ventral sides were carefully examined. The prevalence of macrosymbionts on the starfish *E. luzonicus* was recorded. The most common symbiotic organism on *E. luzonicus* was the ectoparasitic snail *Melanella martinii* (A. Adams in Sowerby, 1854), followed by the pontoniine shrimp *Zenopontonia soror* (Nobili, 1904) and the rare polychaete scaleworm *Asterophilia carlae* Hanley, 1989. The prevalence ratio with host *E. luzonicus* was low and varied by 8.62% and 4.35%, 6.03% and 0%, and 0.86% and 0.72% in July and September 2020 for *M. martinii*, *Z. soror*, and *A. carlae*, respectively. The present study is the first to discover the scaleworm *A. carlae* as a macrosymbiont of the tropical starfish *E. luzonicus*, with a widespread distribution, off Taiwan's northeastern coast, an area influenced by the Kuroshio Current.

## Introduction

Kueishan Island (also known as Gueishan or Turtle Island) is a tiny active volcanic island located off the northeastern coast of Taiwan [1,2]. Shallow hydrothermal vents are located on the east and southeast sides of the island, which features a low pH and a high sulfur concentration [3,4]. This toxic environment has low biodiversity [5]; only a few species of mollusks [5,6], crabs [2,7], benthic copepods [8,9], and cnidarians [5,10] have been recorded. By contrast, the hydrothermal vents slightly affect the seawater on the northwest side of the island. The coastal area northwest of Kueishan Island has a coral reef zone with a length of approximately 150 m and a depth range of 1–8 m. Several biological studies have been conducted in

**Funding:** Financial support from the Ministry of Science and Technology (MOST) of Taiwan through grant no. MOST 108-2811-M-019-504, MOST 109-2811-M-019-504 and MOST 110-2811-M-019-504 to L.-C. Tseng, as well as grant no. MOST 106-2621-M-019-001, MOST 107-2621-M-019-001, MOST 108-2621-M-019-003, MOST 109-2621-M-019-002, MOST 110-2621-M-019-001 and MOST 111-2621-M-019-001, and Center of Excellence for Ocean Engineering (Grant No. 109J13801-51 110J13801-51, 111J13801-51) to J.-S. Hwang. The funders had no role in study design, data collection and analysis, decision to publish, or preparation of the manuscript.

**Competing interests:** The authors declare that they have no conflict of interest.

this small and healthy reef zone; for example, Hung [11] reported 256 fish species belonging to 42 family, and Limviriyakul [12] recorded 57 species of symbiotic decapods from various hosts, including algae, sponges, hydroids, actiniarians, scleractinians, alcyonarians, crinoids, and echinoids.

Echinoderms, a well-defined and highly-derived clade of metazoans with approximately 7,000 species, can be found in various habitats ranging from shallow intertidal areas to abyssal depths [13]. Numerous echinoderms have been found with diverse macrosymbiotic organisms, including feather stars (crinoids) [14–16], sea cucumbers (reviewed by Martin & Britayev [17]; Purcell et al. [18]), sea urchins [15,16,19–21], brittle stars [17], and starfish [17,19], as well as the small crustacean copepod associated with brittle stars [22]. Approximately 1,500 species of starfish live in all marine waters [23], and 48 valid species from 10 families have been recorded in waters around Taiwan [24,25]. Notably, in the reef and coastal waters of Taiwan, several studies have investigated symbiotic shrimps [15,26–28] and symbiotic crabs [20,22,29,30] on various host creatures. However, the evidence on the symbionts of echinoderms in the waters of Taiwan is scant [15,20], and no reports on the symbionts, particularly macrosymbionts, of starfish in the waters of Taiwan are available. Therefore, the diversity of starfish symbionts remains understudied.

Baseline information on the macrosymbionts of echinoderms is required. Thus, during the investigation program of faunal diversity in the present study, the often encountered starfish *Echinaster luzonicus* (Gray, 1840) in the reef zone of northwest Kueishan Island was studied to determine the diverse macrosymbionts of *E. luzonicus*, and the prevalence ratio of each macrosymbiont of *E. luzonicus* was compared.

## Materials & methods

### Study area description

Kueishan Island, with an area of approximately 2.841 km$^2$, is an active volcanic island in the vicinity of Yilan City off Taiwan's northeastern coast, in the southeastern East China Sea (Fig 1). Kueishan Island is so named for its turtle-shaped topography. Shallow hydrothermal vents can be found on the eastern and southeastern sides of Kueishan Island. The western side faces eastern Taiwan, and the northwestern coast has a shallow band zone of coral reefs of approximately 150 m in length (Fig 2A). Numerous fish species inhabit this small and healthy reef area (Fig 2B). The seabed is sandy in the deeper waters.

### Field sampling and sample treatment

Investigations of faunal diversity were undertaken in July and September 2020. Starfish sampling in the 150-m coral reef zone was undertaken at a depth of 8 m through scuba diving. For each type of potential macrosymbiont, both the dorsal and ventral sides were carefully examined. The prevalence of macrosymbionts on the starfish *E. luzonicus* was recorded. In the study area, *E. luzonicus* was often encountered in the reef zone at depths between 5 and 8 m, with polymorphisms of morphology and color. Specifically, colors ranged from bright orange to dark brown, and most individuals had six arms with a length of approximately 5–8 cm. Some starfish were comet shaped, with one bigger arm capable of regenerating a disc with small arms (Fig 2C). Unexpectedly, a scaleworm *A. carlae* was found (Fig 2D).

Only a few symbiont individuals that were found for the first time were placed separately in a plastic ziplock bag and transported to the laboratory at National Taiwan Ocean University for photography and species identification.

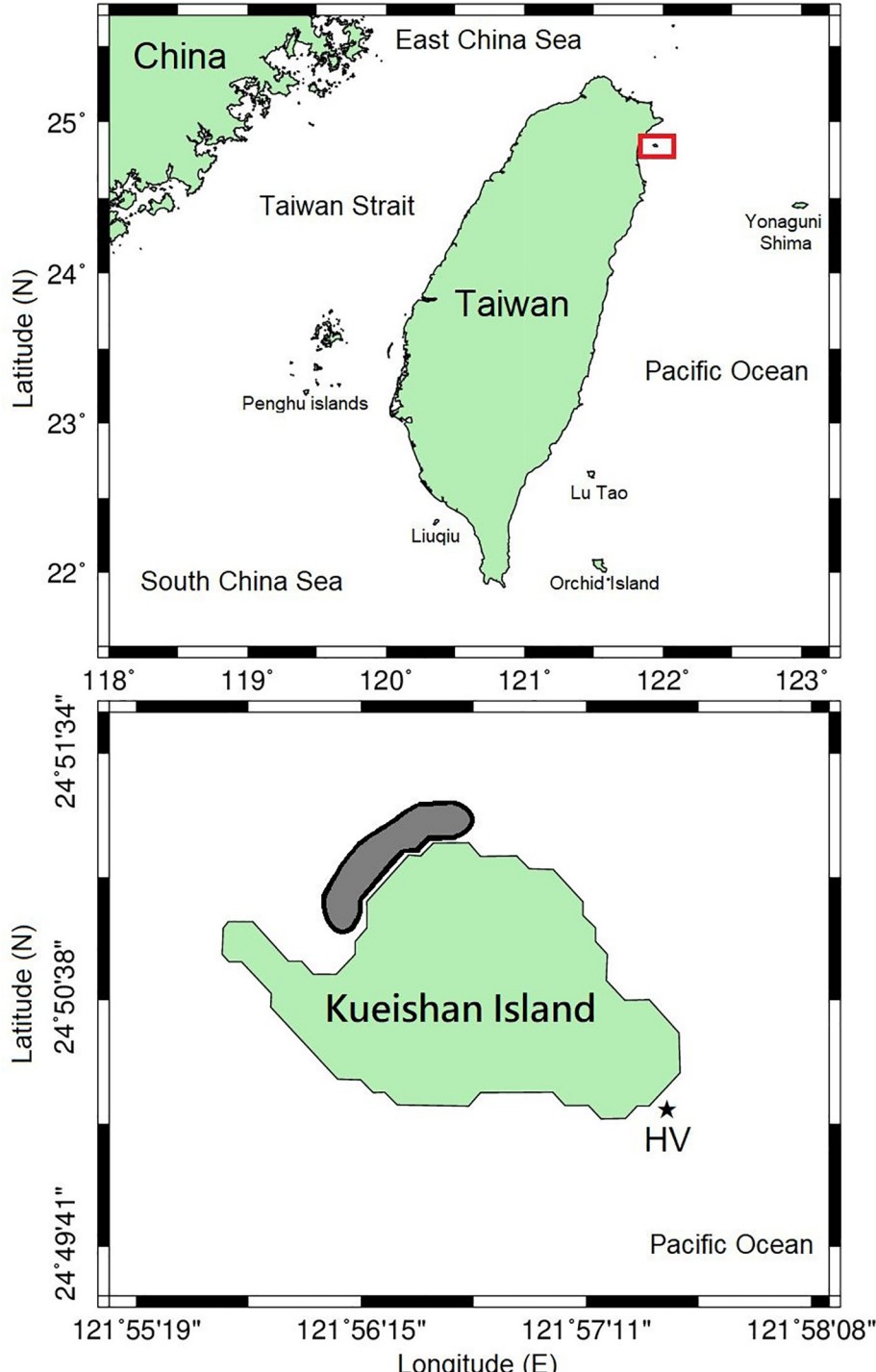

**Fig 1. Map of the study area (upper); the gray area indicates the sampling locations (lower) around Kueishan Island from July to September 2020.** The asterisk indicates the hydrothermal vent (HV) area.

## Sample identification

In the laboratory, the collected animals were identified under a dissecting microscope (Olympus SZX16, Tokyo, Japan) using the keys described by the following researchers: Fauchald

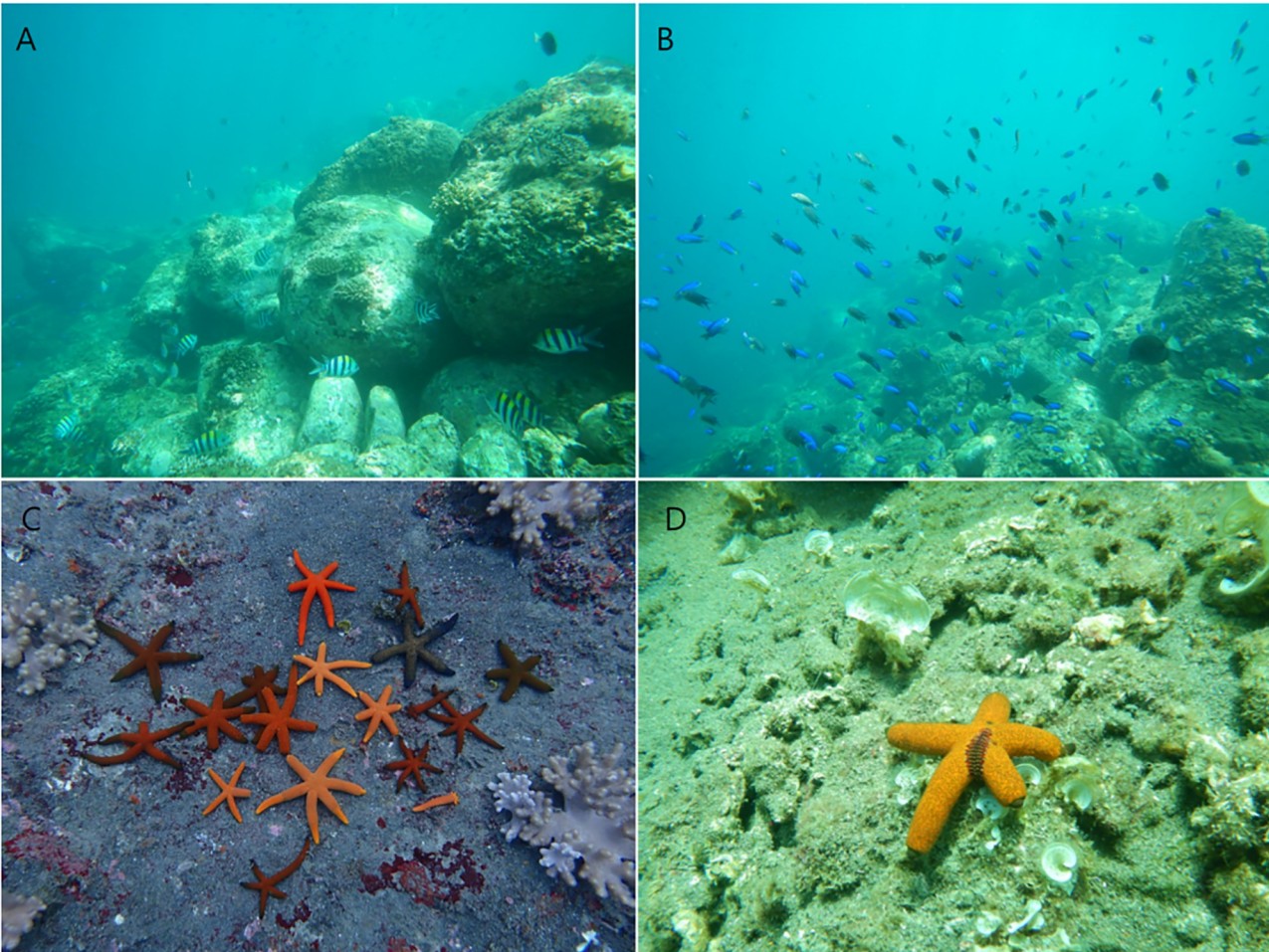

**Fig 2.** The reef zone of northwestern Kueishan Island (A). The small and healthy reef has diverse fish (B), the starfish *Echinaster luzonicus* has highly variable colors (C), and the scaleworm *Asterophilia carlae* can be found in the field (D).

[31], Hanley [32], and Britayev and Fauchald [33] for polychaetes; Adams [34] and Okutani [35] for sea snails; and Bruce [36], Chace and Bruce [37], and Holthuis [38] for shrimp.

## Results

### Macrosymbionts of *E. luzonicus*

A total of 116 and 138 starfish (including the comet-shaped ones) were found in July and September 2020, respectively. On the northwest side of Kueishan Island, three species of macrosymbionts belonging to three classes of the animal kingdom were found on *E. luzonicus*, as follows: Gastropoda, *Melanella martinii* (A. Adams in Sowerby, 1854) (Littorinimorpha: Eulimidae) (Fig 3); Malacostraca, *Zenopontonia soror* (Nobili, 1904) (Decapoda: Palaemonidae) (Fig 4); and Polychaeta, *Asterophilia carlae* Hanley, 1989 (Phyllodocida: Polynoidae) (Fig 5) (Table 1). The scaleworm *A. carlae* found was the first discovery of this species in waters around Taiwan.

The three macrosymbiont species were found in different locations on the external body of the starfish. The snail *M. martinii* was found on the ventral side and was attached to the podia (also known as tube feet) close to the mouth or under the arms. The shrimp *Z. soror* was

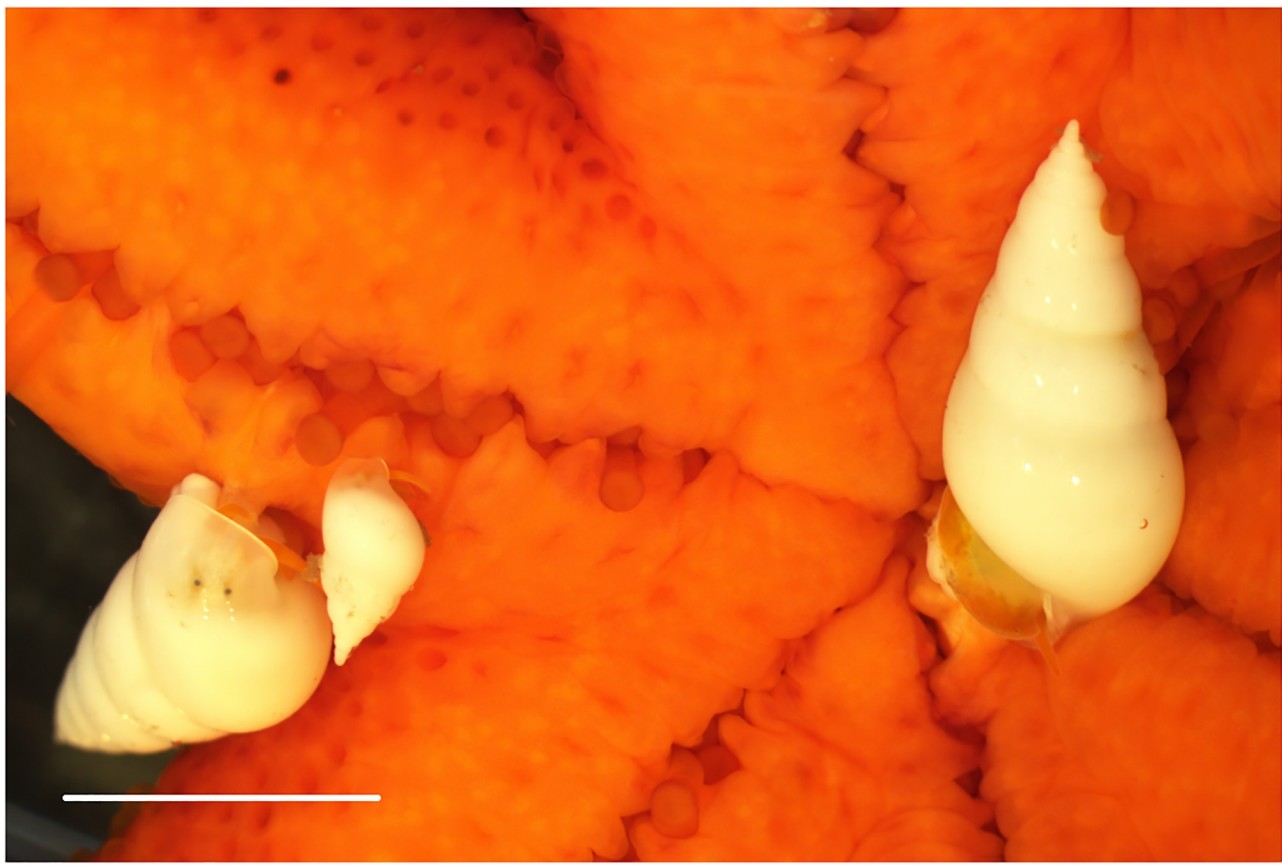

**Fig 3. Symbiotic gastropod: White parasitic snail *Melanella martinii* (C. B. Adams, 1850).** The scale bar = 10 mm.

camouflaged (Fig 4) and was found on the ventral and lateral sides. Occasionally, this species was found underneath the starfish. The polychaete *A. carlae* was found on the ventral and lateral sides and occasionally on the dorsal side of the starfish, and it did not swim away when the host starfish was examined. Notably, all starfish individuals only hosted one species of macrosymbiont, and no more than one macrosymbiont species was observed on the same host starfish in the investigation.

## Prevalence of macrosymbionts

The snail *M. martinii* and the polychaete *A. carlae* were recorded in the July and September 2020 investigations. In July and September 2020, the prevalence of the snail *M. martinii* was 8.52% and 4.35%, and that of the polychaete *A. carlae* was 0.86% and 0.72%, respectively. *Z. soror* only appeared in July, and its prevalence was 6.03% (Table 1). Regarding the number of symbiotic organisms recorded, 1–4, 2–5, and 1 individual snails, shrimps, and polychaetes were observed on starfish, respectively.

## Discussion

### Historical studies of *E. luzonicus*

*Echinaster luzonicus* is widely distributed in the intertidal zones and reefs of the Indo Pacific [39]. Several studies have recorded this species around the South China Sea, including in

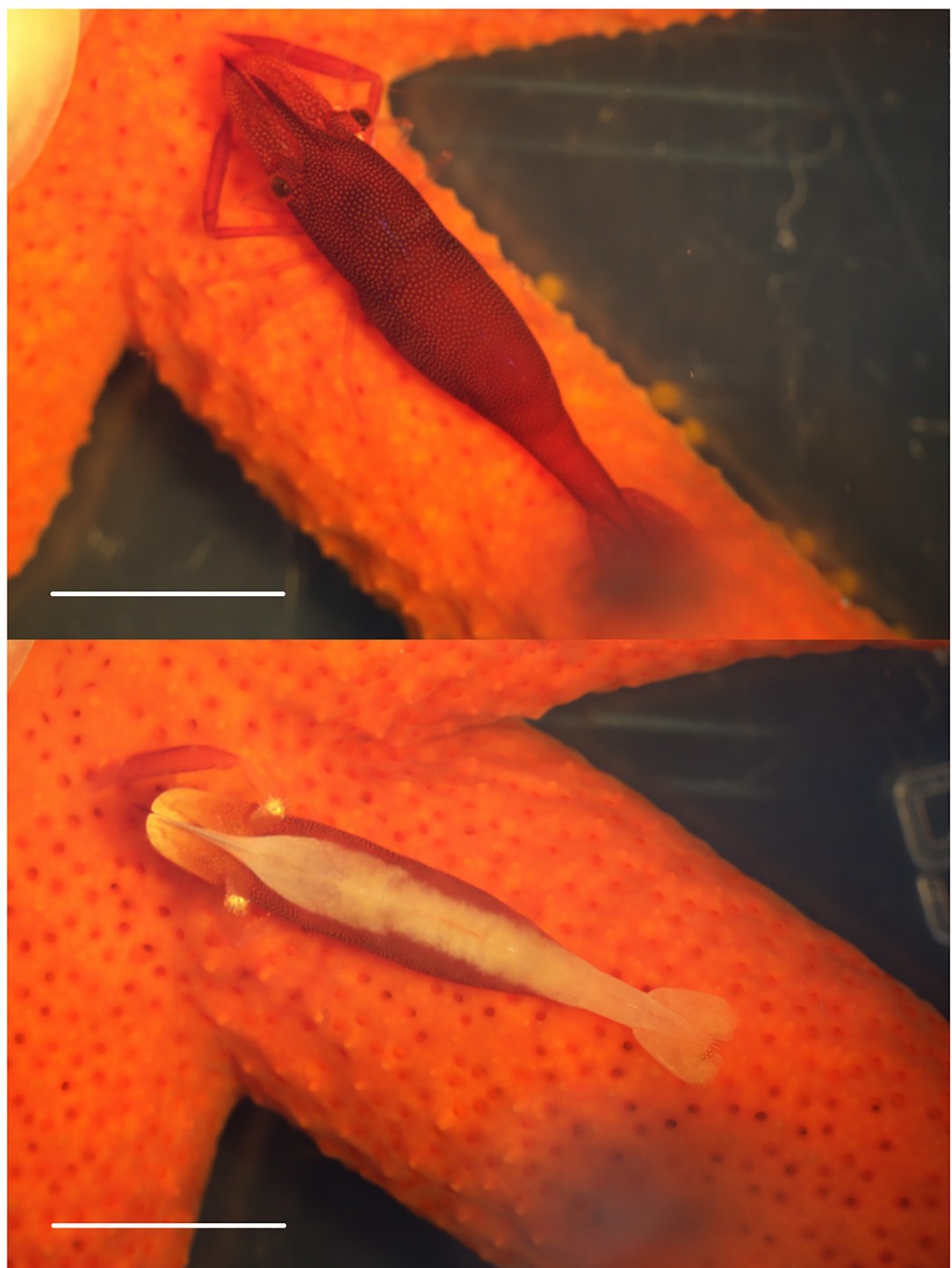

**Fig 4. Symbiotic starfish shrimp: *Zenopontonia soror* (Nobili, 1904) with two types of color patterns.** The scale bar = 5 mm.

Taiwan [24], the Penghu Islands in the Taiwan Strait [40], the Dongsha Atoll of the northern South China Sea [41], southern Vietnam [42], Thailand [43], the Maldives and the Andaman and Nicobar Islands [44], the central South China Sea and Malaysia [45], Taiping Island in the southern part of the South China Sea [46], and Indonesia [47]. As mentioned, in the present

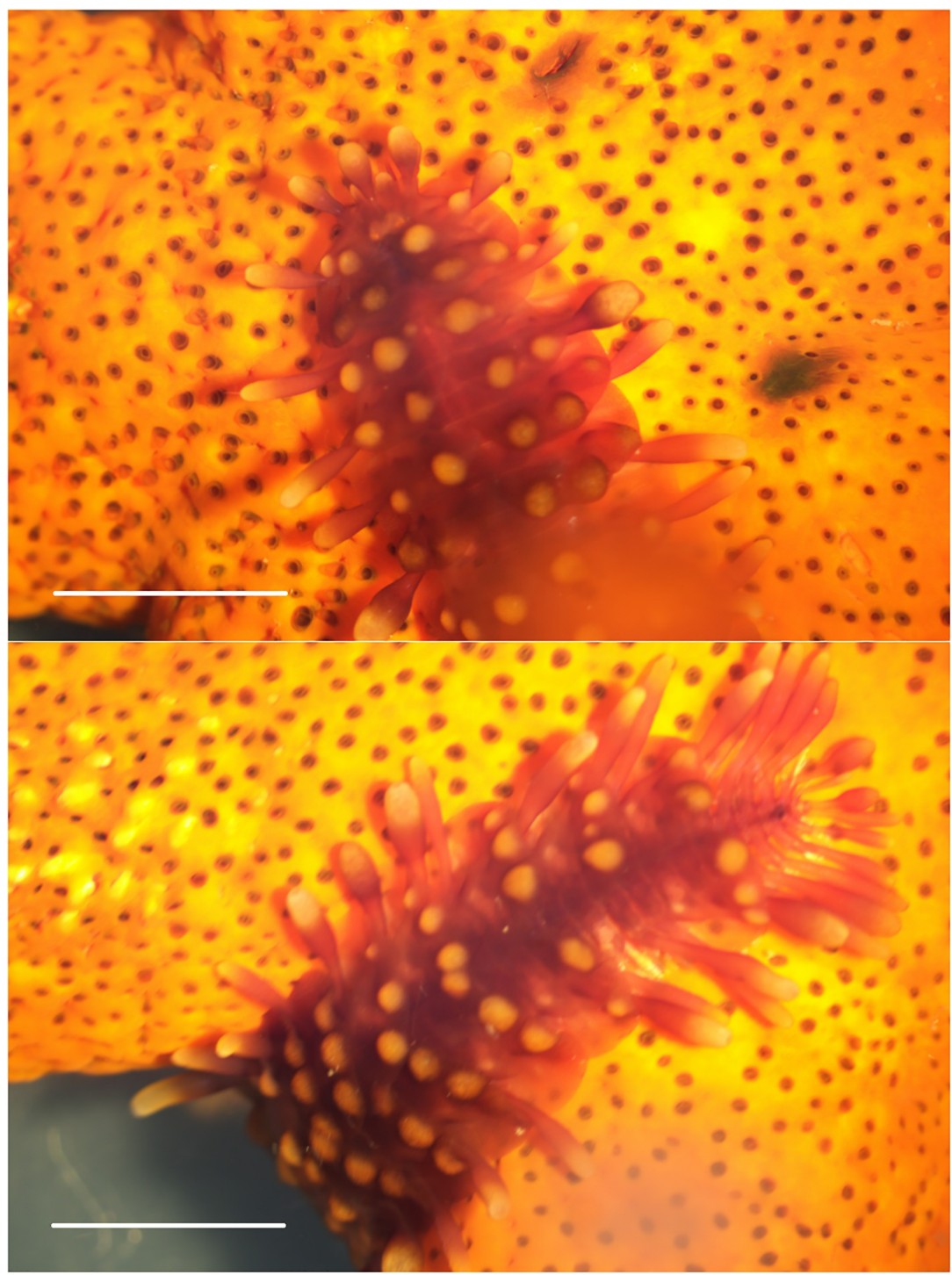

**Fig 5. Symbiotic polychaete:** *Asterophilia carlae* **Hanley, 1989, anterior view (upper) and posterior view (lower).** The scale bar = 5 mm.

**Table 1. Symbionts on starfish *Echinaster luzonicus* and their prevalence (%), as recorded in the July and September 2020 investigations.**

| Symbiont species | Prevalence | | Synonymised names |
|---|---|---|---|
| | July | Sept. | |
| *Melanella martinii* (Gastropoda) | 8.62 | 4.35 | *Eulima martinii* A. Adams in Sowerby, 1854 (MolluscaBase, 2021), *Melanella candida* F.P. Marrat, 1880 (Galli, 2015) |
| *Zenopontonia soror* (Malacostraca) | 6.03 | 0 | *Periclimenes* (*Cristiger*) *frater* Borradaile, 1915, *Periclimenes bicolor* Edmondson, 1935, *Periclimenes frater* Borradaile, *Periclimenes parasiticus* Borradaile, 1898, *Periclimenes soror* Nobili, 1904 (WoRMS, 2021) |
| *Asterophilia carlae* (Polychaeta) | 0.86 | 0.72 | None (Read & Fauchald, 2021b) |

study, the morphology and color of *E. luzonicus* observed varied. Of individuals of this species found in the Indian Ocean, Soota and Sastry [44] reported that they were autotomous, with five or more arms of unequal length. Their diverse color patterns indicate the high genetic variability of *E. luzonicus* within a population, with 22.9% polymorphic loci among 35 genetic loci in samples collected from Ryukyu Islands, Japan [48].

Several studies have reported that different macrosymbionts coexist on *E. luzonicus*, including the scaleworm *Asterophylia culcitae* [42], ctenophoran *Coeloplana astericola* [42,49,50], and shrimp *Z. soror* [42,51]. The copepods *Doridicola echinasteris* and *Stellicola oreastriphilus* have also been found [42]. They have evolved cryptically colors that are indistinguishable from those of its host starfish [52]. In the present study, three macrosymbionts of *E. luzonicus* were found in a small coral reef area. Until now, *M. martinii* and *A. carlae* had never been reported in the waters around Taiwan.

## Macrosymbionts of *E. luzonicus*

Symbiosis, a mode of interaction between two heterospecific organisms, can be designated into three categories, namely mutualism, commensalism, and parasitism, depending on the presence or absence of "harm" or "benefit" in the partners [53]. In theory, the criteria of host mortality and metabolic dependency have often been used to determine the type of their interactions. However, the methods required to apply such criteria in marine ecosystems are costly and time consuming [54,55]. The type of symbiosis can be determined through indirect methods, such as laboratory observation of feeding behavior, in situ analysis [56] of the morphology and function of the mouth and foraging organ [57–59], analysis of the digestive tract [60], and analysis of intestinal contents [55].

Coral reefs contain the largest diversity of symbiotic associations in marine environments [54]. At least 860 invertebrate species live in close association with stony corals, and they depend on their hosts for food and habitat [61]. These symbiotic associations make coral reef communities the most complex and biodiverse marine ecosystems in coastal areas [61–63]. In reef areas, some host species are strongly reliant on their obligate symbionts to the point where they are unable to survive without them [61,64]. Understanding the modes in which marine fauna interact can help clarify their ecological roles and provide new insights into natural science. In this study, we found three species of macrosymbionts on *E. luzonicus*: *M. martinii*, *Z. soror*, and *A. carlae*. Several studies have revealed various ecological niches in the symbiotic relationship of these species with various hosts.

*Melanella martinii*—Notably, the snail *M. martinii* identified in both months of the investigation had not been reported previously in the waters of Kueishan Island. Studies have found *M. martinii* in the East China Sea, South China Sea, and Indo–West Pacific [65]; Japan [35];

Vietnam [66]; Cebu, Philippines [67]; Singapore [68]; Lombok, Indonesia [69]; and Australia [70]. Several reports have indicated the presence of several mollusks in the shallow waters of hydrothermal vent areas on the eastern side of Kueishan Island, but not *M. martinii* [5,6,71].

Among holothurian symbionts, sea snails in the genus *Melanella* are parasites [18,72–74]. Most of them attach to the hosts' skin, piercing through the tissue with their specialized proboscis and feeding on coelomocytes. These attachment strategies do not have severe effects on the hosts [18,72]. However, in this study, *M. martinii* was associated with starfish, an unusual host in the genus *Melanella*. Thus, due to the shortage of records and relevant evidence, whether the relationship between *M. martinii* and *E. luzonicus* is parasitic in nature remains unclear.

Symbiosis of several mollusks with various starfish species has been reported [50], such as *Thyca crystallina* and *Stilifer* cf. *linckiae* with the blue starfish *Linckia laevigata* [75], *Granulithyca nardoafrianti* with *Nardoa frianti*, *T. crystallina* with *L. laevigata*, and *Stilifer* spp. with *L. laevigata* and *Culcita novaeguineae* in the waters of southern Vietnam [42]. Furthermore, numerous species of starfish are prey and parasitic hosts for gastropods. For example, the giant triton *Charonia tritonis* preys on several asteroids, including the crown-of-thorns starfish *Acanthaster planci* [76]. The mollusks of *Stylifer* spp. consume the tissue of host starfish and are occasionally found within it [42].

Some eulimids are host-specific. A particular genus of eulimids tends to be restricted to a single class level or a lower taxon of echinoderm [72]. Gastropods of the genus *Melanella* are associated with holothurian hosts [72,73]. The present data revealed that *M. martinii* lived on the starfish of the class Asteroidea instead of on the usual holothurian hosts of its congeneric species. The association information of this snail is scarce; thus, further research is warranted. We also observed that the attachment of this snail to the starfish podia caused the starfish no injury. Therefore, we postulate that *M. martinii* might suck the body fluid of starfish but not consume their tissue.

*Zenopontonia soror*–*Z. soror* has a worldwide distribution; it has been found in Hong Kong, Taiwan, Xisha Islands, Hainan, and the Indo-Pacific [65]; southern Taiwan, northern South China Sea [77]; Japan, Taiwan, the Philippines, Indonesia, Papua New Guinea, Australia, New Caledonia, French Polynesia [12]; Vietnam [42,78]; Thailand [51]; and the Colombian Pacific [79]. In the current study, *Z. soror* was only found in the July investigation. Limviriyakul [12] did not find this species among the symbiotic crustaceans collected in the same area in April and September 2015. This shrimp may hide in the crevices of coral reefs or leap swiftly away when it detects an approaching diver [51]. It may also switch from one individual or species to another [78].

The shrimps *Z. soror* has been found on at least 23 species of shallow-water tropical starfish, such as *Acanthaster planci*, *Culcita novaeguineae*, *Choriaster granulatus*, and *Linckia laevigata* in Vietnam [78], as well as on various hosts, such as starfish and cushion stars. These include the crown-of-thorns starfish in Thailand [51], cushion star *Culcita novaeguineae* [80], and starfish *Pentaceraster cumingi* [79]. *Zenopontonia soror* can recognize, differentiate, and obtain protection from host species based on host-provided chemical and visual cues [78,80]. A record high of 25 *Z. soror* individuals was found on a single cushion star in the waters around Ko Waen Island in southern Thailand [51], and an even more astonishing 53 individuals were found on a crown-of-thorns starfish in Kuroshima Island, Japan [81]. The highest record in the present study was five shrimp on a single *E. luzonicus*. The difference in numbers may be attributable to variations in the size of the starfish species. *Echinaster luzonicus* is smaller than the cushion star and the crown-of-thorns starfish; thus, the load capacity of symbiotic shrimp is limited. This is also supported by Antokhina and Britayev [78], who suggested that the distribution of the starfish shrimp on its hosts in Vietnam does not depend directly on host abundance but is rather related to the host size, oral surface area, and morphological complexity.

Clear evidence obtained by directly observing, conducting experiments, and analyzing feeding appendages has led to the argument that *Z. soror* is an obligate commensal symbiont of starfish [36,78,80]. The shrimp highly depends on its host for nutrition and protection. The morphology of the mandible and chelae indicate that *Z. soror* may browse on mucus or mucus-entrapped particles [36]. Olliff [80] suggested that the shrimp may feed on host ectoparasites, similar to other symbionts living on larger hosts. *Zenopontonia soror* has the ability to change its color patterns to match its hosts in order to increase the survival rate [80–82]. These observations demonstrate coevolution between the starfish host and the *Z. soror* symbiont.

*Asterophilia carlae*—The scaleworm *A. carlae* was a noteworthy discovery in our samples. This scaleworm belongs to the highly diverse family Polynoidae, which contains numerous symbiotic species associated with other marine invertebrates [17,83,84]. *Asterophilia carlae* is distributed in regions of the Pacific Ocean, Fiji, and temperate, subtropical, and tropical waters [85]. Taxonomic records reveal that this species had not been previously recorded in waters around Taiwan or in seas adjacent to Mainland China [65]. This is the first record of *A. carlae* with *E. luzonicus* in Taiwanese waters since it was originally observed on the blue starfish *Linkia laevigater laevigata* in Fijian and Indonesian waters [32]. Its congener species, *A. culcitae* Britayev & Fauchald, 2005 [33] was first reported to be distributed in Vietnam [42], but it has yet to be found in Taiwanese waters. Historical records of *A. carlae* in the waters of Taiwan are not available. The record made in the present study is the northernmost record of this species in the world. Notably, this study is the first to document *A. carlae* in these regional waters. However, details concerning the presence of *A. carlae* in the waters of Taiwan remain unclear.

Hosts of *A. carlae* are mainly restricted to the class Asteroidea; host species in the class Crinoidea are rare [17]. However, the relationship between *A. carlae* and many symbiont polynoid scaleworms and their hosts remains poorly understood [86]. *Gastrolepidia clavigera*, a widespread polynoid scaleworm, is very similar to *A. carlae* and is ectosymbiotic with holothuroids [33]. This species was considered commensal [17] until Britayev and Lyskin [55] revealed it to be a parasite; it was found to feed on host tissue. It also feeds on parasitic copepods, but this does not afford greater advantage than does association with a host. Nevertheless, some symbiotic relationships between two partners can be shifted depending on the situation, such as from commensalism to parasitism [87]. Knowledge of *A. carlae* and its hosts is insufficient to determine the mode of symbiosis. Further studies should carefully account for the presence or absence of "harm" and "benefit" to reveal the true interspecific interaction.

The intrusion of warm South China Sea water through the Luzon Strait in the northern South China Sea may play a pivotal role in transporting the copepod *Calanoides philippinensis* to the northern waters of the western Pacific Ocean under the influence of the Kuroshio Current [88,89]. The results suggest that *A. carlae* in planktonic larval stage may also use the same pathway to move northward to the Kueishan Island reef area. In sum, the Kuroshio Current might contribute crucially to the geographic dispersal and distribution of *A. carlae*.

Martin and Britayev [17] reported a prevalence of 3.3%–13% but did not mention the examined number of hosts. The prevalence might be influenced by variable bathymetric [90], spatial [91], temporal [17], and host-related factors [92]. The present study examination of 116 and 138 starfish species in July and September 2020, respectively, demonstrated that the prevalence of *A. carlae* was low (average: 0.79%). We confirm the presence of *A. carlae* in the study area. Further studies are warranted to provide information on species population, seasonal succession, and other biological factors.

## Conclusion

Among diverse heterospecific associations, the best visualized results may be obtained for symbiosis, which is maintained through spatiotemporal adaptive interactions [53]. The present study documented three species of ectosymbionts obtained from *E. luzonicus*. The findings provide insights into the relationship between each macrosymbiont species and *E. luzonicus*; furthermore, they advance the knowledge of the ecological role of starfish and their symbiotic associations. Determining whether the relationship of these macrosymbionts with their host starfish was epibiotic, commensal, or parasitic was challenging. We found no evidence of injuries on the surface or soft tissue of host starfish. Studies on intraspecific interactions of symbionts and their possible effects on the growth of reef starfish under laboratory conditions are required to gain a comprehensive understanding of their symbiotic relationships in nature.

## Supporting information

**S1 File. Supporting information file provides information of all figures.**
(XLSX)

## Acknowledgments

The authors would like to thank anonymous reviewers for their invaluable, detailed suggestions and criticisms, which greatly enhanced this paper.

## Author Contributions

**Conceptualization:** Li-Chun Tseng, Jiang-Shiou Hwang.

**Data curation:** Li-Chun Tseng, Parinya Limviriyakul.

**Formal analysis:** Li-Chun Tseng.

**Funding acquisition:** Jiang-Shiou Hwang.

**Investigation:** Li-Chun Tseng.

**Methodology:** Li-Chun Tseng, Parinya Limviriyakul.

**Project administration:** Jiang-Shiou Hwang.

**Resources:** Jiang-Shiou Hwang.

**Software:** Li-Chun Tseng, Jiang-Shiou Hwang.

**Supervision:** Jiang-Shiou Hwang.

**Validation:** Li-Chun Tseng, Parinya Limviriyakul.

**Visualization:** Li-Chun Tseng.

**Writing – original draft:** Li-Chun Tseng, Parinya Limviriyakul.

**Writing – review & editing:** Li-Chun Tseng, Parinya Limviriyakul, Jiang-Shiou Hwang.

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
