## [Decision Letter · Decision Letter 0]

18 Jul 2022

PONE-D-21-26592Macrosymbionts of starfish Echinaster luzonicus (Gray, 1840), with a new biogeographic record of scaleworm Asterophilia carlae Hanley, 1989, in the waters of a volcanic western Pacific islandPLOS ONE

Dear Dr. Hwang,

Thank you for submitting your manuscript to PLOS ONE. After careful consideration, we feel that it has merit but does not fully meet PLOS ONE’s publication criteria as it currently stands. Therefore, we invite you to submit a revised version of the manuscript that addresses the points raised during the review process.

1. Data in the current manuscript are from the year 2020. More data are needed for this work. 

2. The "Discussion" part needs in-depth-discussion.

3. This manuscript needs  a suitable title.

4. All figures should be added the scale bar.

If you plan to submit a revision, the revision will be sent to previous reviewers for further consideration. Please submit your revised manuscript by Sep 01 2022 11:59PM. If you will need more time than this to complete your revisions, please reply to this message or contact the journal office at plosone@plos.org. Please include the following items when submitting your revised manuscript:A rebuttal letter that responds to each point raised by the academic editor and reviewer(s). You should upload this letter as a separate file labeled 'Response to Reviewers'.A marked-up copy of your manuscript that highlights changes made to the original version. You should upload this as a separate file labeled 'Revised Manuscript with Track Changes'.An unmarked version of your revised paper without tracked changes. You should upload this as a separate file labeled 'Manuscript'.

We look forward to receiving your revised manuscript.

Kind regards,

Wan-Xi Yang, Ph.D.

Academic Editor

PLOS ONE

Journal Requirements:

3. Thank you for stating the following financial disclosure: "Financial support from the Ministry of Science and Technology (MOST) of Taiwan through grant no. MOST 108-2811-M-019-504, MOST 109-2811-M-019-504 and MOST 110-2811-M-019-504 to L.-C. Tseng, as well as grant no. MOST 106-2621-M-019-001, MOST 107-2621-M-019-001, MOST 108-2621-M-019-003, MOST 109-2621-M-019-002, and MOST 110-2621-M-019-001, and Center of Excellence for Ocean Engineering (Grant No. 109J13801-51) to J.-S. Hwang. The funders had no role in study design, data collection and analysis, decision to publish, or preparation of the manuscript."

We note that one or more of the authors is affiliated with the funding organization, indicating the funder may have had some role in the design, data collection, analysis or preparation of your manuscript for publication; in other words, the funder played an indirect role through the participation of the co-authors. If the funding organization did not play a role in the study design, data collection and analysis, decision to publish, or preparation of the manuscript and only provided financial support in the form of authors' salaries and/or research materials, please do the following:

a. Review your statements relating to the author contributions, and ensure you have specifically and accurately indicated the role(s) that these authors had in your study. These amendments should be made in the online form.

b. Confirm in your cover letter that you agree with the following statement, and we will change the online submission form on your behalf: 

“The funder provided support in the form of salaries for authors [insert relevant initials], but did not have any additional role in the study design, data collection and analysis, decision to publish, or preparation of the manuscript. The specific roles of these authors are articulated in the ‘author contributions’ section.

5. Please upload a copy of Supporting Information S1 file which you refer to in your text on page 24.

Reviewers' comments:

Reviewer's Responses to Questions

**Comments to the Author**

1. Is the manuscript technically sound, and do the data support the conclusions?

Reviewer #1: Yes

Reviewer #2: Partly

2. Has the statistical analysis been performed appropriately and rigorously? 

Reviewer #1: Yes

Reviewer #2: No

3. Have the authors made all data underlying the findings in their manuscript fully available?

Reviewer #1: Yes

Reviewer #2: Yes

4. Is the manuscript presented in an intelligible fashion and written in standard English?

Reviewer #1: Yes

Reviewer #2: Yes

5. Review Comments to the Author

Reviewer #1: After reading the whole manuscript, I do not see the reasons to why scaleworm Asterophilia carlae should appear in the title as there are other new findings as well. What is so special with scaleworm Asterophilia carlae?

Results were well presented and discussed. However, I suggest discussion to start with general observation followed by specific data. Therefore, from lines 277 to 290 where the others are discussing general observation, should have stated earlier.

Line 155: I don’t know the reason why the others present issues of medicinal value of the species Echinaster luzonicus. I think the others were supposed to confine themselves in distribution and variability among the study species especially in this paragraph.

Compare lines 157-159 and lines 130-132: While previous studies show different macrosymbionts to coexist on E. luzonicus, the findings of this study shows each individuat starfish to host one species of macrosymbiont. I therefore expected this to be discussed here

Reviewer #2: The data of the ms are only from 2020, so the results are somewhat accidental and simple, the reliability need to be considered. It is recommended to increase the sampling time. All figures should be added the scale bar.

6. PLOS authors have the option to publish the peer review history of their article (what does this mean?). If published, this will include your full peer review and any attached files.

Reviewer #1: No

Reviewer #2: No

---

## [Author Response · Author response to Decision Letter 0]

4 Sep 2022

Author responses to referee comments (29th August 2022)

Date: Jul 18 2022 03:12PM

To: "Jiang-Shiou Hwang" jshwang@mail.ntou.edu.tw;jshwang@ntou.edu.tw

From: "PLOS ONE" plosone@plos.org

Subject: PLOS ONE Decision: Revision required [PONE-D-21-26592]

PONE-D-21-26592

Macrosymbionts of starfish Echinaster luzonicus (Gray, 1840), with a new biogeographic record of scaleworm Asterophilia carlae Hanley, 1989, in the waters of a volcanic western Pacific island

PLOS ONE

Dear Dr. Hwang,

Thank you for submitting your manuscript to PLOS ONE. After careful consideration, we feel that it has merit but does not fully meet PLOS ONE’s publication criteria as it currently stands. Therefore, we invite you to submit a revised version of the manuscript that addresses the points raised during the review process.

1. Data in the current manuscript are from the year 2020. More data are needed for this work.

2. The "Discussion" part needs in-depth-discussion.

3. This manuscript needs a suitable title.

4. All figures should be added the scale bar.

If you plan to submit a revision, the revision will be sent to previous reviewers for further consideration.

We look forward to receiving your revised manuscript.

Kind regards,

Wan-Xi Yang, Ph.D.

Academic Editor

PLOS ONE

Authors’ response: We are thankful for the very constructive comments and efforts from two reviewers. We fully agree with the suggestions and have corrected them accordingly and hopefully eradicated errors.

Journal Requirements:

3. Thank you for stating the following financial disclosure: "Financial support from the Ministry of Science and Technology (MOST) of Taiwan through grant no. MOST 108-2811-M-019-504, MOST 109-2811-M-019-504 and MOST 110-2811-M-019-504 to L.-C. Tseng, as well as grant no. MOST 106-2621-M-019-001, MOST 107-2621-M-019-001, MOST 108-2621-M-019-003, MOST 109-2621-M-019-002, and MOST 110-2621-M-019-001, and Center of Excellence for Ocean Engineering (Grant No. 109J13801-51) to J.-S. Hwang. The funders had no role in study design, data collection and analysis, decision to publish, or preparation of the manuscript."

We note that one or more of the authors is affiliated with the funding organization, indicating the funder may have had some role in the design, data collection, analysis or preparation of your manuscript for publication; in other words, the funder played an indirect role through the participation of the co-authors. If the funding organization did not play a role in the study design, data collection and analysis, decision to publish, or preparation of the manuscript and only provided financial support in the form of authors' salaries and/or research materials, please do the following:

a. Review your statements relating to the author contributions, and ensure you have specifically and accurately indicated the role(s) that these authors had in your study. These amendments should be made in the online form.

b. Confirm in your cover letter that you agree with the following statement, and we will change the online submission form on your behalf:

“The funder provided support in the form of salaries for authors [insert relevant initials], but did not have any additional role in the study design, data collection and analysis, decision to publish, or preparation of the manuscript. The specific roles of these authors are articulated in the ‘author contributions’ section.

5. Please upload a copy of Supporting Information S1 file which you refer to in your text on page 24.

Review Comments to the Author

Reviewer #1: 

After reading the whole manuscript, I do not see the reasons to why scaleworm Asterophilia carlae should appear in the title as there are other new findings as well. What is so special with scaleworm Asterophilia carlae?

Authors’ response: We are thankful for the constructive comments. Since scaleworm Asterophilia carlae was found for the first time in the study waters, therefore its name was specifically put in the title. We changed the title to “Macrosymbionts of starfish Echinaster luzonicus (Gray, 1840) in the waters of a volcanic western Pacific island”.

Results were well presented and discussed. However, I suggest discussion to start with general observation followed by specific data. Therefore, from lines 277 to 290 where the others are discussing general observation, should have stated earlier.

Authors’ response: We are thankful for the very constructive comments and efforts from the reviewer. In the corrected version, we have rearranged the paragraphs and descriptions as suggested.

Line 155: I don’t know the reason why the others present issues of medicinal value of the species Echinaster luzonicus. I think the others were supposed to confine themselves in distribution and variability among the study species especially in this paragraph.

Authors’ response: Thanks for the useful advice. This sentence has been deleted as suggested.

Compare lines 157-159 and lines 130-132: While previous studies show different macrosymbionts to coexist on E. luzonicus, the findings of this study shows each individuat starfish to host one species of macrosymbiont. I therefore expected this to be discussed here.

Authors’ response: The term "coexist" means that these symbiotic animals have all been recorded on starfish E. luzonicus. These references do not record the presence of two coexisting animals on a single starfish individual. 

Reviewer #2: 

The data of the ms are only from 2020, so the results are somewhat accidental and simple, the reliability need to be considered. It is recommended to increase the sampling time. All figures should be added the scale bar.

Authors’ response: We are thankful for the suggestions. Due to research funding constraints, we can only conduct the survey in 2020, so we do not have survey data for 2021 to the present. The survey will continue in the future with the support of funding. In July of this year (2022), we visited the surveyed reefs again and only recorded Zenopontonia soror symbiosis with starfish E. luzonicus. So far, there are very few records of Asterophilia carlae, and probably its population is not abundant, so the chance of discovery is also very low. In 2020, both surveys covered more than 90% of the reef area, and the results obtained can definitely be relied on. We have added a scale bar in Figures 3-5 in the corrected version of the manuscript.

---

## [Decision Letter · Decision Letter 1]

15 Nov 2022

Macrosymbionts of starfish Echinaster luzonicus (Gray, 1840), with a new biogeographic record of scaleworm Asterophilia carlae Hanley, 1989, in the waters of a volcanic western Pacific island

PONE-D-21-26592R1

Dear Dr. Hwang,

We’re pleased to inform you that your manuscript has been judged scientifically suitable for publication and will be formally accepted for publication once it meets all outstanding technical requirements.

Kind regards,

Wan-Xi Yang, Ph.D.

Academic Editor

PLOS ONE

Additional Editor Comments (optional):

Reviewers' comments:

Reviewer's Responses to Questions

**Comments to the Author**

1. If the authors have adequately addressed your comments raised in a previous round of review and you feel that this manuscript is now acceptable for publication, you may indicate that here to bypass the “Comments to the Author” section, enter your conflict of interest statement in the “Confidential to Editor” section, and submit your "Accept" recommendation.

Reviewer #1: All comments have been addressed

Reviewer #3: All comments have been addressed

2. Is the manuscript technically sound, and do the data support the conclusions?

Reviewer #1: Yes

Reviewer #3: Yes

3. Has the statistical analysis been performed appropriately and rigorously? 

Reviewer #1: Yes

Reviewer #3: Yes

4. Have the authors made all data underlying the findings in their manuscript fully available?

Reviewer #1: Yes

Reviewer #3: Yes

5. Is the manuscript presented in an intelligible fashion and written in standard English?

Reviewer #1: Yes

Reviewer #3: Yes

6. Review Comments to the Author

Reviewer #1: I accept the manuscript after major revision done. The authors were able to answer one question after another I gave in my first review. They also improved presentation of the manuscript, i find it to be better than the way it was before. Basing on my understanding and the way they discussed the data, I recommend that the manuscript is accepted

Reviewer #3: (No Response)

7. PLOS authors have the option to publish the peer review history of their article (what does this mean?). If published, this will include your full peer review and any attached files.

Reviewer #1: No

Reviewer #3: **Yes: **Boping Tang

---

## [Editor Report · Acceptance letter]

18 Nov 2022

PONE-D-21-26592R1 

Macrosymbionts of starfish *Echinaster luzonicus* (Gray, 1840) in the waters of a volcanic western Pacific island 

Dear Dr. Hwang:

I'm pleased to inform you that your manuscript has been deemed suitable for publication in PLOS ONE. Congratulations! Your manuscript is now with our production department. 

Kind regards, 

on behalf of

Dr. Wan-Xi Yang 

Academic Editor

PLOS ONE